

# Technical note: Quadratic solution of the approximate reservoir equation (QuaSoARe)

Julien Lerat [1]

[1]CSIRO Environment, Canberra, ACT, 2601, Australia

*Correspondence to*: Julien Lerat (julien.lerat@csiro.au)

**Abstract.** This paper presents a method to solve the reservoir equation, a special type of scalar ordinary differential equation controlling the dynamic of conceptual reservoirs found in most hydrological models. The method called "Quadratic Solution of the Approximate Reservoir Equation" (QuaSoARe) is applicable to any reservoir equation regardless of its non-linearity or the number of fluxes entering and leaving the reservoir. The method is based on a piecewise quadratic interpolation of the flux functions, which lead to an analytical and mass conservative solution. It is applied to two routing models and two rainfall-runoff stores that are representatives of hydrological model components and evaluated on six catchments located in Eastern Australia that experienced one of the most extreme floods in recent Australian history. A comparison of the method against two standard numerical schemes, the Radau fifth order implicit and Runge-Kutta of order 5(4) explicit schemes suggests that it can reach similar accuracy while reducing runtime by a factor of 10 to 50 depending on the model considered. At the same time, the model code is simple enough to be presented as a short pseudo-code included in our paper. Beyond solving a given reservoir equation, the method constitutes a promising avenue to define flexible models where flux functions are defined as piecewise quadratic functions, which can be solved exactly with QuaSoARe.

## 1   Introduction

### 1.1   Reservoirs as ubiquitous components of environmental models

Environmental models, including hydrological models often rely on components that can be modelled conceptually as a reservoir receiving inputs and generating outputs that are sole functions of the volume stored in the reservoir. Such reservoirs are extensively used in rainfall-runoff models such as GR4J (Perrin et al., 2003), HBV (Bergstrom and Forsman, 1973), IHACRES (Croke and Jakeman, 2004) and SAC-SMA (Burnash and Ferral, 1981). This paper presents an approximate analytical method called Quadratic Solution of the Approximate Reservoir Equation (QuaSoARe) to solve the scalar

ordinary differential equation (ODE) underlying most conceptual reservoirs used in hydrology.

The solution of ODE represents an entire field in applied mathematics with an history as old as differential calculus. The reader might wonder why a paper is needed on such a beaten scientific track. Despite the voluminous literature written on the topic, the large number of software packages available and the importance of this topic flagged by Kavetski and Clark





(2010), the use of well-proven ODE numerical schemes remains rare in hydrological modelling. Our explanation for this

troubling fact is that, perhaps, the well-established methods described in reference text books (Hairer et al., 2009; Shampine, 2020) aim at solving very general ODE and hence require complex algorithms to handle every extreme scenario in the equation setup. However, this complexity might be superfluous if one wants to solve a simple scalar equation such as conceptual reservoirs used in hydrological models. This paper aims at bridging this gap by proposing a simple, yet effective and mass conservative, approximate solution specifically designed for the scalar reservoir equation ODE.

More precisely, if the reservoir volume is denoted by $S$, the reservoir equation ODE is formulated as

$$\frac{dS}{dt} = \sum_{i=1}^{n} f_i(S, \tilde{V}) \qquad \text{Eq. 1}$$

where $f_i$ are arbitrary continuous functions representing input or output fluxes from the reservoir and $\tilde{V}$ is a vector containing forcing variables (e.g., rainfall or potential evapotranspiration). Here we assume that the $f_i$ are Lipschitz continuous which ensures that Eq. 1 has a unique solution (Hairer et al., 2009). We also assume that Eq. 1 is input-to-state stable which guarantees that its solution is bounded if the inputs $\tilde{V}$ are bounded (Mironchenko, 2023). Note that demonstrating the global

stability of an ODE is a complex problem much beyond the scope of this paper. Methods using Lyapunov functions are presented by LaSalle (1960) and generalized by Sontag (1989) for systems such as Eq. 1.

Solving the reservoir equation in Eq. 1 is an initial value problem over a time interval $[0, \delta]$ where $\delta$ is the time step and $S_0$ is the initial condition at $t = 0$. This process is subsequently repeated for each time step of the simulation with varying forcing variables (e.g., a daily series of rainfall values). It is highlighted that the presence of several functions $f_i$ in Eq. 1 is

common in hydrological models, for example to account for multiple runoff generation processes such as infiltration and overland flow or fluxes between surface water and groundwater stores (Clark et al., 2008). These fluxes are often required by other components of the hydrological model (e.g. infiltration excess runoff being routed by a routing model) which means that, in addition to solving for variable $S$, one must compute the total of each flux over the time step given by

$$O_i = \int_0^{\delta} f_i(S, \tilde{V}) dt \qquad \text{Eq. 2}$$

Eq. 1 has a broad range of applications beyond hydrological modelling, for example to estimate storage in an artificial

reservoir (Fiorentini and Orlandini, 2013) or to solve the gradually varied flow equation in hydraulics (Gill, 1976). Unfortunately, there are very few cases where both Eq. 1 and Eq. 2 have an analytical solution, a problem that is considerably more difficult than solving Eq. 1 alone. Consequently, most reservoir equations are solved using numerical approximation methods.



## 1.2    Numerical methods to solve the reservoir equation

The most common approach relies on discrete methods that estimate $S$ from $t = 0$ to $\delta$ at incremental steps using Runge-Kutta methods (Kavetski and Clark, 2010; Knoben et al., 2019; La Follette et al., 2021). These schemes are the topic of a voluminous literature including several reference books (Butcher, 2003; Hairer et al., 2009; Press et al., 2007; Shampine, 2020 to name but a few). Nonetheless, applying them requires a significant expertise to (1) select the most appropriate scheme among the multiple variants available (for example Euler, Runge-Kutta, or Huen schemes) (2) decide if the scheme

is explicit (when the solution depends on $S_0$ only) or implicit (if it depends on both $S_0$ and $S(\delta)$, which requires iterative optimization), (3) choose between fixed or variable step length to automatically slow down computation when facing numerical difficulties. These choices are not trivial and far from harmless as warned by Michel et al. (2003) and Kavetski and Clark (2011) who show the disastrous consequences of solving the exponential store with an explicit Euler scheme. As a result, these techniques are not simple to code and a modeler who is not familiar with them will often require a third-party

software package. This adds a dependency to the model code, complicates maintenance and increases runtime, sometimes significantly when using implicit methods, compared to an analytical solution. At the same time, despite the exponential growth of computing power, runtime is still a limiting factor in hydrology when a large number of runs is required, for example in a Monte Carlo uncertainty analysis or for the calibration of distributed models. Finally, jointly solving Eq. 1 and Eq. 2 using a numerical solver requires transforming the scalar equation Eq. 1 to a system of differential equations by adding

one scalar equation for each flux:

$$\frac{dO_i}{dt} = f_i(S, \tilde{V}) \quad i = 1, \dots n \qquad \textbf{Eq. 3}$$

Because of the wide range of magnitudes observed in hydrological fluxes, this may lead to a system of equations that could be characterized as "stiff" for which explicit numerical schemes can become unstable (Kavetski and Clark, 2011; Shampine, 2020).

Another angle of attack for solving an ODE is to replace the original equation by one for which an analytical solution exists.

The simplest ordinary differential equations being linear, it is not surprising that linearization of Eq. 1 around a certain regime, e.g. steady state, has constituted the first approach proposed by mathematicians (Hartman, 2002). Linearization has often been used to solve complex differential equations in hydrology and hydraulics such as the Saint-Venant 1D hydrodynamic equation, for example by Hayami (1951) or more recently by Fan and Li (2006) or Munier et al. (2008). This approach is efficient if the true solution does not depart significantly from the linearization regime. Unfortunately,

hydrological systems often exhibit variations of several orders of magnitude that violate this assumption. A logical extension of the linearization approach is to define several linear approximation regimes between which the solution can switch. This idea has been explored extensively in the control literature (Johansson, 2003) starting from early work by Kalman (1955). More recently, this approach has been formalized in the field of Electrical Engineering under the name "Trajectory Piecewise-Linear Approximation" (TPLA), a method introduced by Rewienski and White (2003) to solve large non-linear





differential equation systems. The theory presented by Rewienski and White (2003) along with its subsequent refinements (Bond and Daniel, 2009; Kalra and Nabi, 2020) aims to solve equation systems that are far more complex than the reservoir equation studied here. Consequently, an adaptation of this theory for the scalar reservoir equation along with a clear algorithmic description would be a useful contribution from our side. In addition, the TPLA theory relies on a transition between linearized states which is not necessarily continuous. Finally, the choice of the transition function is arbitrary, which

adds complexity to the process, and does not guarantee that the derivative of the solution remains continuous. This can be an issue if the model shows strong non-linearity. This problem was raised by Litrico et al. (2010) who proposed a piecewise-linear model to approximate the Saint-Venant hydro-dynamic equations. Overall, the TPLA method is useful for representing complex non-linear dynamics, but a simpler approach restricted to a scalar system such as the reservoir equation would likely make its adoption easier. Extending the linearisation idea, Pope (1963) introduced the concept of exponential

integrator where the linearisation of an ODE is combined with a discrete method such as a Runge-Kutta scheme applied to the residual between the linearised part and the original function. Hochbruck and Ostermann (2006) demonstrated the efficacy of this method to solve large system of stiff equations. However, its reliance on a discrete method to correct the linearised solution faces the same issues raised earlier for an application to hydrological models.

Overall, the review of the literature above highlighted the following research gaps:

• The reservoir equation is a common tool in hydrology which requires fast and robust numerical solutions in the absence of general analytical approaches.
      • Classical ODE numerical solvers such as Runge-Kutta methods are not straightforward to use, especially if the equations are stiff and lead to potentially unstable solutions.
      • Existing theoretical developments from control theory based on multiple linearization points are complex and
require care when switching between linearized regimes.

### 1.3   Objectives of the paper

This paper aims at

      • Presenting an approximate analytical solution for the reservoir equation. The solution solves Eq. 1 and computes all input and output fluxes from Eq. 2.
• Demonstrating the application of the method to four reservoir equations of increasing complexity and non-linearity and compare the results with classical implicit and explicit discrete methods.

The method, including its pseudo-code, is presented in Section 2.2 and 2.3 while an accompanying python package is released as supporting material. Section 4 details some limitations of the method and recommendations to remediate them. The protocol used to compare our method with existing discrete methods is detailed in Section 5 with results presented in

Section 6 and discussed in Section 7. The paper is concluded in Section 8.





## 2 Approximate analytical solution of the reservoir equation using piecewise quadratic functions

The method presented in this section is inspired by the TPLA and exponential integrators methods where the functions $f_i$ in Eq. 1 are approximated by functions for which an analytical solution of the reservoir equation exists.

### 2.1 Illustrative example

Before presenting our method, we introduce an illustrative example using the following reservoir equation:

$$\frac{dS}{dt} = -\frac{S^3}{2} \qquad \textbf{Eq. 4}$$

This equation is a special case of the cubic flow routing model further discussed in Section 5.1 which has an analytical solution $s(t)$ given by given the initial condition $S_0$ at $t = 0$:

$$s(t) = \frac{S_0}{\sqrt{1 + t \, S_0^2}} \qquad \textbf{Eq. 5}$$

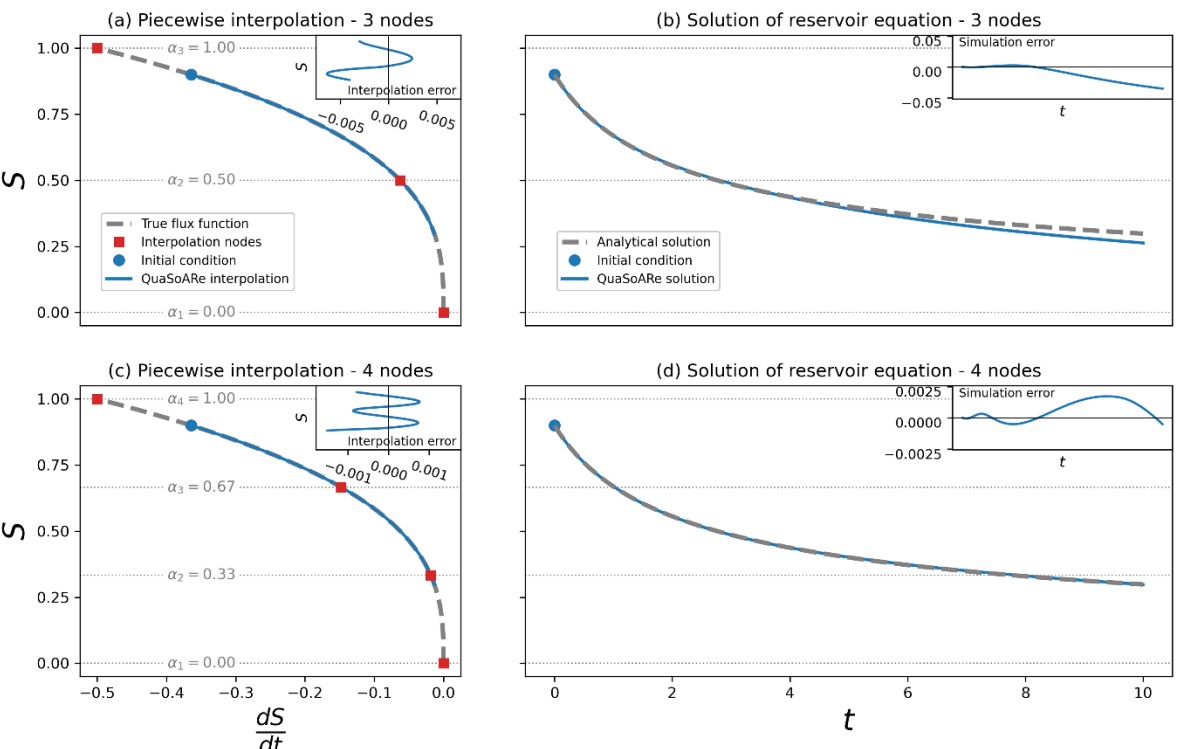

**Figure 1: QuaSoARe method applied to the example reservoir equation. The left plots (a and c) present the reservoir equation as a grey dashed line, i.e. the right-hand side of Eq. 4, and its QuaSoARe approximation as a blue line. The right plots (b and d) show the analytical and QuaSoARe solution of this equation using the same colours. A QuaSoARe configuration with three interpolation nodes is used in the two top plots (a and b) while four nodes are used in the two bottom plots (c and d).**





The reservoir equation function from the right-hand side of Eq. 4 is plotted in Figure 1.a as a grey dashed line. The corresponding solution from Eq. 5 is shown in Figure 1.b as a grey dashed line setting $S_0$ to 0.9. Other elements in this figure are related to the QuaSoARe method and are described in the following sections.

## 2.2   Reservoir equation approximation by quadratic functions

Before presenting our approximate solution, we need to establish that the solution $s(t)$ of Eq. 1 is monotonic, a result that is used repeatedly across this paper. This can be proved by contradiction: let us assume that $s(t)$ is not monotonic. Consequently, $s$ being derivable as a solution of Eq. 1, there exists a time $t_0 > 0$ where the derivative of $s$ changes sign and $ds/dt \neq 0$ for $t > t_0$. Let us now introduce function $s_2$ which is identical to $s$ for $t < t_0$ and which remains constant and equal to $s(t_0)$ for $t \geq t_0$. This function is a solution of Eq. 1 for all $t > 0$ and is distinct from $s$ for $t > t_0$ because its derivative is zero while the derivative of $s$ is not. This is a contradiction with the uniqueness of the solution of Eq. 1 imposed by the fact that $f$ is Lipschitz continuous as stated in the previous section. Consequently, $s$ is monotonic for $t \geq 0$.

Back to solving Eq. 1, let us assume that each function $f_i$ can be approximated by a function $f_i^*$ written as:

**Eq. 6**

$$f_i^*(S, \tilde{V}) = \sum_{j=1}^{m-1} \left[ a_{i,j}\, S^2 + b_{i,j}\, S + c_{i,j} \right] I_j(S)$$

where the series $\alpha_1 < \alpha_2 < \cdots < \alpha_m$ defines a partition of the interval $[\alpha_1, \alpha_m]$ into $m + 1$ intervals referred to as "interpolation bands". In addition, the interval $[\alpha_1, \alpha_m]$ is assumed to contain the bounds of $s(t)$ (solution to Eq. 1 is assumed bounded in Section 1.1). $I_j(x)$ is the indicator function equal to 1 if $x \in [\alpha_j, \alpha_{j+1}[$ and zero elsewhere. The $\alpha_j$ values are referred to as interpolation nodes in this paper. The coefficients $a_{i,j}$, $b_{i,j}$ and $c_{i,j}$ are fixed. The process to obtain $a_{i,j}$, $b_{i,j}$ and $c_{i,j}$ is detailed in Appendix A so that the approximated function $f_i^*$ matches the original function $f_i$ at the nodes $\alpha_j$ and at a mid-point between the $\alpha_j$.

The approximation described above is applied to our example equation given in Eq. 4: two approximations are presented in Figure 1.a and in Figure 1.c using three and four nodes, respectively. Both lead to a highly accurate interpolation where true and approximated function are visually indistinguishable. The interpolation error, i.e. the difference between the true function and its approximated counterpart, is shown as inset in both plots. This error reduces by a factor of approximately five when increasing the number of nodes from three to four.

The form of Eq. 6 was chosen for several reasons. First, a piecewise quadratic function is Lipschitz continuous on a bounded interval. As a result, the solution to the corresponding reservoir equation, referred to as "approximated reservoir equation", is unique like the one of Eq. 1. Second, this equation can be solved analytically as is shown in Section 2.3. Third, a quadratic function can approximate a wide range of reservoir functions used in hydrology. For example, if $a_{i,j} = 0$, the equation becomes a linear function of $S$ which is the most common reservoir equation used in hydrology. Finally, the steady state solution of the approximate reservoir equation (i.e. when the derivative of $S$ is zero) can be determined analytically, which greatly facilitates the analysis of the behaviour of the reservoir and the selection of the $\alpha_j$ as discussed in Section 3.





Unfortunately, despite these appealing attributes, there are also downsides of this approximation that are presented in Section
4 along with potential remediation.

Replacing the original functions $f_i$ by their approximated counterpart $f_i^*$, the approximate reservoir equation is

$$\frac{dS}{dt} = \sum_{i=1}^{n} f_i^*(S, \tilde{V}) = \sum_{j=1}^{m} \left[A_j\, S^2 + B_j\, S + C_j\right] I_j(S) \qquad \textbf{Eq. 7}$$

where $A_j$, $B_j$ and $C_j$ are the sums of the corresponding coefficients for interpolation band $j$. For example, $A_j$ is given by

$$A_j = \sum_{i=1}^{n} a_{i,j} \qquad \textbf{Eq. 8}$$

Importantly, Eq. 7 maintains an equality between the change in storage and the sum of fluxes which means that it conserves
mass. This statement is trivial, but it ensures that the QuaSoARe algorithm generates mass conservative simulations, which
is a key requirements in hydrological modelling and is not always guaranteed by ODE numerical schemes as pointed out by
Clark and Kavetski (2010).

## 2.3    Analytical solution of the approximated reservoir equation

The solution to Eq. 7 can be obtained analytically as follows. Let us assume that the initial condition $S_0$ is falling in the
interpolation band $[\alpha_{j_0}, \alpha_{j_0+1}[$ where $j_0 \in [1, m-1]$. For example, $j_0 = 2$ (second band) if $S_0 = 0.9$ in our illustrative
example shown in Figure 1.a. When $S_0$ falls into the $j_0^{th}$ interpolation band at $t = 0$, Eq. 7 can be simplified as

$$\frac{dS}{dt} = A_{j_0}\, S^2 + B_{j_0} S + C_{j_0} \qquad \textbf{Eq. 9}$$

The solution of this equation has an analytical expression, referred to as $s(t)$. In addition, all fluxes from Eq. 2 can also be
computed analytically. The process to obtain these expressions is not complex but tedious so its presentation is deferred to
Appendix B. Using solution $s(t)$, one can compute the value $s(\delta)$ at the end of the time step, leading to three cases:

- **Case 1**: $s(\delta)$ lies in the interval $[\alpha_{j_0}, \alpha_{j_0+1}[$ : because it is monotonic (see beginning of Section 1.1), $s(t)$ remains
bounded by $\alpha_{j_0}$ and $\alpha_{j_0+1}$ for all $t$ in $[0, \delta]$. Consequently, Eq. 7 remains identical to Eq. 9 for $t \in [0, \delta]$ which
        means that $s(t)$ is the solution of equation Eq. 7 over the interval $[0, \delta]$.

- **Case 2**: $s(\delta) < \alpha_{j_0}$ : here, $s(t)$ is a continuous function with $s(0) = S_0 > \alpha_{j_0}$ and $s(\delta) < \alpha_{j_0}$. Hence, by the
        intermediate value theorem, there exists a time $t_l < \delta$ such that $s(t_l) = \alpha_{j_0}$. Moreover, $s(t)$ is decreasing which
        means that $s(t)$ remains in the interval $[\alpha_{j_0}, S_0]$ for all $t < t_l$. In other words, $s(t)$ is the solution of Eq. 7 for $t$ in
$[0, t_l]$. The expression for $t_l$ has an analytical expression equal to $v(\alpha_{j_0}) - v(S_0)$ where function $v$ is given in Eq.
        23 (see Appendix B).





When $t > t_l$, Eq. 7 is no longer equivalent to Eq. 9 because $s(t)$ becomes lower than $\alpha_{j_0}$. As a result, $s(t)$ is no longer the solution of Eq. 7. However, Eq. 7 becomes equivalent to an equation similar to Eq. 9 where $A_{j_0}$ and $B_{j_0}$ are replaced by $A_{j_0-1}$ and $B_{j_0-1}$, respectively.

- **Case 3**: $s(\delta) > \alpha_{j_0+1}$ : following a similar reasoning than above, $s(t)$ is a strictly increasing function until it reaches the value $\alpha_{j_0+1}$ at time $t_u < \delta$ where $t_u$ is equal to $\nu(\alpha_{j_0+1}) - \nu(S_0)$ and $s$ is the solution of Eq. 7 for $t \in [0, t_u]$.

  For $t > t_u$, $s$ becomes greater than $\alpha_{j_0+1}$ which means that it is no longer the solution of Eq. 7. However, Eq. 7 then becomes equivalent to an equation similar to Eq. 9 where $A_{j_0}$ and $B_{j_0}$ are replaced by $A_{j_0+1}$ and $B_{j_0+1}$, respectively. .

Overall, the first case above leads to an immediate resolution of Eq. 7 over the interval $[0, \delta]$ while the other cases provide a solution over two shorter intervals $[0, t_l]$ and $[0, t_u]$ corresponding to cases 2 and 3, respectively. For these last two cases, if $t$ is greater than $t_u$ or $t_l$, Eq. 7 becomes equivalent to an equation similar to Eq. 9 where the index $j_0$ is replaced by either $j_0 - 1$ or $j_0 + 1$, which suggests that the whole process can be repeated iteratively towards the end of the interval $t = \delta$.

It is worth mentioning that the value $s(\delta)$ in the above algorithm may not be defined. It is possible for certain values of the coefficients $A_j, B_j, C_j$ to lead to a solution $s(t)$ becoming infinite before reaching $t = \delta$. The last column of Table 3 in Appendix B indicates the time interval during which $s(t)$ remains valid depending on these coefficients. This situation obviously excludes Case 1 above but can be captured under Case 2 or 3 if the invalid value $s(\delta)$ is replaced by $+\infty$ or $-\infty$ if $s$ is increasing or decreasing, respectively. Note that this adjustment of the algorithm ensures that it can cope with an invalidity of $s(\delta)$, but does not guarantee its systematic convergence. This question is discussed further in Section 4.

It is important to highlight that each step in the algorithm described above is explicit because the calculation of $s(\delta)$ depends on values of past values of $S$ only. In addition, the underlying analytical solution (see Appendix B) is computed using a limited set of standard mathematical functions (exponential, logarithm, hyperbolic tangent and tangent). Both elements suggest that the QuaSoARe method is simple and fast to implement in any programming language. Finally, each QuaSoARe iteration is repeated at most $m + 1$ times if the entire range $[\alpha_1, \alpha_m]$ is traversed by the solution. Consequently, the runtime required to compute the approximate solution is bounded by the number of interpolation nodes and cannot reach high values like what can happen with algorithms relying on variable time step size such as classical implementation of Runge-Kutta methods.

QuaSoARe is applied to our illustrative example with the approximated solution shown in Figure 1.b (interpolation using three nodes) and Figure 1.c (four nodes) in as blue lines. QuaSoARe solutions closely match the analytical solution (dashed grey line) for both interpolation configurations. However, the match degrades in Figure 1.b towards the end of the simulation with QuaSoARe leading to an underestimation of the true solution. This is due to the negative interpolation error when $S$ is lower than 0.5 seen in the inset of Figure 1.a which leads to a more rapid decrease of the approximated solution and hence





progressive underestimation. This example highlights the potential of QuaSoARe to generate accurate simulations, but also the need for high accuracy in the interpolation of flux functions.

In summary, the solution presented in this section provides a way to solve an approximate reservoir equation where the reservoir functions are replaced by piecewise quadratic interpolations. The solution is fully analytical including the computation of all reservoir fluxes. It can be implemented using the pseudo code presented in Figure 2. Alternatively, Lerat (2024) released an open-source software package where QuaSoARe is coded in both Python and C languages.





```
QuaSoARe(α, A, B, C, t₀, S₀, δ)
 1    // α ∈ ℝᵐ : interpolation nodes
 2    // a, b, c ∈ ℝᵐ⁻¹ˣⁿ : interpolation coefficients for each band and flux
 3    // t₀: start time
 4    // S₀: initial condition
 5    // δ : time step
 6
 7    // Initialisation
 t₁ = t₀
 O ∈ ℝⁿ = 0 // fluxes set to 0
j = arg max_k (α[k] < S₀) // Find j₀
while t₁ ≤ δ
13        // Sum of interpolation coefficients across fluxes
A_j = ∑ⁿ_{i=1} a[i, j]
B_j = ∑ⁿ_{i=1} b[i, j]
C_j = ∑ⁿ_{i=1} c[i, j]
18        // Bounds of interpolation band
S⁻ = α[j]
S⁺ = α[j + 1]
22        // Tentative solution at the end of the time step
23        // (Table 3, second column)
24        s(δ) = S(A_j, B_j, C_j, S₀, δ)
25
26        // Move iteration forward in time
if s(δ) ∈ [S⁻, S⁺[
28            // Solution stays in the current interpolation band
t₁ = δ
S₁ = s(δ)
else
32            // Solution leaves the current interpolation band
if s(δ) < S⁻
j* = j − 1
S* = S⁻
else
j* = j + 1
S* = S⁺
40            // Find time when solution leaves the band
41            // (Table 3, first column)
t₁ = t₀ + ν(A_j, B_j, C_j, S*) − ν(A_j, B_j, C_j, S₀)
44            // Update band number
j = j*
47        // Increment fluxes (Equation 29, Appendix B)
for i = 1 to n
O[i] = O[i] + Oᵢ*(t₀, t₁)
51        // Loop
S₀ = S₁
t₀ = t₁
return S₁, O
```

**Figure 2: QuaSoARe pseudo-code.**



## 3  Selecting interpolation nodes from the range of steady state solutions

The QuaSoARe method presented in Section 2 relies on an interpolation of functions $f_i$ using nodes $\{\alpha_j\}_{j=1,\dots,m}$. The choice

of these nodes is important because it conditions the quality of the interpolation and hence of the approximate solution. Let

us now assume that the reservoir equation is solved for a fixed time step $\delta$ and a sequence of $p$ forcing values $\{\tilde{V}_k\}_{k=1,\dots p}$. In

most reservoirs used in hydrology, the reservoir equation has a set of steady state solutions for each time step $k$, noted $\bar{S}_{k,l}$,

that are solutions of

$$\sum_{i=1}^{n} f_i(\bar{S}_{k,l}, \tilde{V}_k) = 0 \qquad \textbf{Eq. 10}$$

If the set of steady state solutions $\{\bar{S}_{k,l}\}$ is not empty, a simple approach is to set $\alpha_1$ to $\min(\{\bar{S}_{k,l}\})$ and $\alpha_m$ to $\max(\{\bar{S}_{k,l}\})$.

Subsequently, the $\alpha_i$ are obtained by partitioning the interval $[\alpha_1, \alpha_m]$ into $m-1$ equal sub-intervals. This approach relies

on the fact that extreme values of $s(t)$ are reached during time steps where the forcings $\tilde{V}_k$ are likely to be extremes. At the

same time, steady state solutions are storage values that are reached when the integration time step $\delta$ tends to infinity.

Consequently, when an extreme forcing value is used, a simulation run for an infinite time step is likely to result in storage

values that are higher than any other values of $s(t)$ seen when using a finite time step, and hence constitute a conservative

estimate for their bounds.

In addition, the steady state solutions, and hence their bounds, are straightforward to compute with QuaSoARe because, if

they exist, they are the roots of a quadratic polynomial (right-hand side of Eq. 9) which can be computed analytically from

the coefficients $A_j$, $B_j$ and $C_j$. The corresponding functionality is included in the QuaSoARe software package (Lerat, 2024).

Unfortunately, the existence of steady-state solutions is not guaranteed for all reservoir equations. For example the

exponential reservoir with zero inflows does not have one (Michel et al., 2003). If this the case, we recommend using a trial-

and-error approach to obtain a set of $\alpha_j$ that covers the entire range of $s(t)$ observed during the simulation.

## 4  Limitations of the method and recommendations

The QuaSoARe method is designed to solve a scalar ordinary differential equation. Hence, it cannot be used to solve a

coupled system of equations. This is an important limitation as most hydrological models contain multiple stores that could

benefit from a joint solution as pointed out by Clark and Kavetski (2010). Extension of QuaSoARe to higher dimensions is

not straightforward because the analytical solutions underpinning the method do not have a vector equivalent. However, in

the case where the model reservoirs operate in sequence with no feedback, a simple solution consists in applying QuaSoARe

to each reservoir in turn at a finer time interval than the desired time step, generating the fluxes over each sub-interval and

using these results to feed the next reservoir in the model structure. This is arguably less efficient from a runtime perspective

than applying QuaSoARe over the whole time step, but probably not dissimilar to discrete methods that often shorten the

time step to very short sub-steps to control the error.





The second limitation comes from the quality of the quadratic interpolation underlying QuaSoARe. Figure 1 clearly shows that small discrepancies between the true and interpolated flux functions can lead to noticeable simulation errors. The solution to this problem is to increase the number of interpolation nodes as was shown for our illustrative example in Figure 1 where the interpolation errors reduce by a factor of five when switching from three to four nodes. This is straightforward to implement if a modeller starts with a high number of nodes leading to an interpolation error smaller than her machine precision as is done in Section 5. Such a small error level is theoretically achievable as the reservoir functions are assumed continuous, and hence can be approximated up to any error level by a piecewise polynomial. If this configuration exceeds the modellers' runtime requirement, then the number of nodes is progressively reduced to match this constraint.

Particular care should be taken with the interpolation of the reservoir function close to steady state values discussed in the previous section. Functions with sharp transitions (e.g. rational fractions) cannot be interpolated accurately by a quadratic polynomial over large intervals. Consequently, if applied without constraint, the piecewise interpolation can overshoot and create erroneous steady state values that are not existing in the original equation (i.e. values of $S$ where $f_i^*(S, \tilde{V})$ is null but not $f_i(S, \tilde{V})$). To avoid this problem, a constraint is imposed in the computation of the interpolation coefficients presented in Appendix A to restrict the quadratic functions to be monotonic and prevent them to cross the 0 line if the original flux function did not.

Despite the recommendations above, QuaSoARe, like all numerical ODE solvers, is not guaranteed to converge for all reservoir equations and initial conditions. In particular, reservoir equations often define regions of stability where solutions starting from similar initial condition remain close for all $t > 0$. If the interpolation accuracy is low, it is possible that the approximate and original stability regions do not coincide. As a result, certain values of the initial condition may lead to a stable solution for the original equation but not for QuaSoARe. Fortunately, this problem is related to interpolation accuracy and can be diagnosed before running QuaSoARe. Similarly to what is mentioned above, it is recommended to run QuaSoARe using a high number of interpolation nodes first to ensure that stability regions are approximated accurately. Ultimately, the validity interval of the analytical solutions can be verified at runtime using expressions given in Table 3. Consequently, it is possible to catch this type of problem before attempting a QuaSoARe iteration. Note that this case was never encountered in practice while running the tests presented in the following section where QuaSoARe is applied to a range of non-linear reservoir models using challenging hydro-climate data.

## 5 Comparison of the method with alternative numerical schemes

### 5.1 Reservoir equations tested

The QuaSoARe method is applied to four reservoir equations presented in Table 1 representing common hydrological models of increasing complexity and non-linearity. All the equations in this table depend on a storage scaling factor $\theta$



expressed in the same unit as $S$. This factor acts like a storage capacity and controls the dynamic of the reservoir response. Ten values of $\theta$ are evaluated for each equation using bounds for $\theta$ indicated in the last column of Table 1.

The first two equations simulate the routing of an inflow time series from the upstream to downstream end of a river reach.
This type of routing model is used in semi-distributed hydrological or flood forecasting models to approximate the solution of hydrodynamic equations (Hapuarachchi et al., 2022). The reach receives an inflow, noted $Q_{inflow}(t)$, assumed fixed over the time step $t$. The ouflow from the reach is computed as a power function of the reach storage following Yevdjevitch (1959) where $\beta$ is the power exponent and $Q_{ref}$ is a constant reference flow introduced to simplify dimensional analysis (see Table 1). The routing model is run at an hourly time step. The case where $\beta = 1$ corresponds to the linear routing model
(Meyer, 1941) while $\beta = 2$ is the quadratic routing store solved analytically by Bentura and Michel (1997). Both cases can be treated as quadratic functions of the storage and hence be solved exactly with QuaSoARe. To provide a more meaningful challenge for our method, the two cases where $\beta = 3$ (cubic reservoir, CR) and $\beta = 6$ (bi-cubic) reservoirs are selected.

The third equation underlies the production store of the GR4J model (Perrin et al., 2003). This model is a well-established daily conceptual rainfall-runoff model used worldwide. Its production store, referred to as GR, receives rainfall ($P$) and
potential evapotranspiration ($E$) and generates three fluxes: infiltrated rainfall in the store, actual evapotranspiration from the store and percolation leaked from the store. The first two fluxes are quadratic functions of the store level, which could be solved exactly with QuaSoARe, but the percolation flux introduces a strong non-linearity with a fifth order polynomial. Note that the version of the GR store used in this paper computes the three fluxes simultaneously in a single equation similarly to Santos et al. (2018) whereas they are solved sequentially using the operator splitting method in the original version of the
GR4J model presented by Perrin et al. (2003).

The fourth equation, noted GRM (GR modified), is inspired by the GR equation but increases the non-linearity of the fluxes by converting the infiltrated rainfall and the actual evapotranspiration to fifth order polynomials. In addition, a fourth flux is added to represent groundwater recharge in the form of a rational fraction. Rational fractions are difficult to interpolate with polynomials as they possess asymptotes, which leads to a challenging case for QuaSoARe.

We highlight that the main objective of introducing these four reservoir equations is to obtain challenging tests for QuaSoARe that are representative of hydrological models in use. The objective is not to improve existing models such as GR4J or reproduce accurately observed data from existing catchments. Consequently, one should not be surprised by the unusual formulation of certain equations in Table 1.




**Table 1: Reservoir equations used for the testing of QuaSoARe**

| Name | Description | Fluxes | $\theta$ parameter range |
|---|---|---|---|
| CR | Cubic and Bi-Cubic routing models<br><br>Hourly time step | Inflow: $f_1(S) = Q_{inflow}$<br><br>Outflow: $f_2(S) = -Q_{ref}\,(S/\theta)^3$ | $[1800\,Q_{ref},\ 18000\,Q_{ref}]$ |
| BCR | Bi-Cubic routing models<br><br>Hourly time step | Inflow: $f_1(S) = Q_{inflow}$<br><br>Outflow: $f_2(S) = -Q_{ref}\,(S/\theta)^6$ | $[1800\,Q_{ref},\ 18000\,Q_{ref}]$ |
| GR | GR4J production store<br><br>Daily time step | Infiltrated rainfall: $f_1(S) = P[1 - (S/\theta)^2]$<br><br>Actual evapotranspiration:<br>$f_2(S) = -E\,(S/\theta)(2 - S/\theta)$<br><br>Percolation: $f_3(S) = -\frac{2.25^4}{4}\theta\,(S/\theta)^5$ | $[100, 1000]$ |
| GRM | Modified GR4J production store including groundwater recharge<br><br>Daily time step | Infiltrated rainfall:<br>$f_1(S) =$<br>$\qquad P\,[1 - (S/\theta)^3(10 - 15\,S/\theta + 6(S/\theta)^2)]$<br><br>Actual evapotranspiration:<br>$f_2(S) = -E\left[16\left(S/\theta - \frac{1}{2}\right)^5 + \frac{1}{2}\right]$<br><br>Percolation: $f_3(S) = -\frac{2.25^4}{4}\theta\,(S/\theta)^7$<br><br>Groundwater recharge:<br>$f_4(S) = -0.1\,S/\theta/(1 + 10\,S/\theta)$ | $[100, 1000]$ |

## 5.2 Performance evaluation

The solution of the four reservoir equations is computed with QuaSoARe using 5, 50 and 500 interpolation nodes. These three configurations lead to an increasing accuracy of the interpolation of flux functions, which is expected to translate into higher accuracy of the solution.

QuaSoARe is compared with two discrete numerical schemes: the Radau IIA implicit Runge-Kutta method of order 5 (Hairer and Wanner, 1996), referred to as "Radau", and the explicit Runge-Kutta method of order 5(4) (Dormand and Prince,

1980), noted "RK45". The Radau method was chosen because it is implicit (hence able to handle stiff equations) and of high order, both characteristics leading us to qualify its outputs as the reference for comparison with other methods. The RK45



method was selected because it is the de facto explicit ODE solver that is widely recognised for its combination of speed and accuracy (Shampine, 2020). Both methods are run using the Scipy package implementation where these two algorithms are coded in Python (Virtanen et al., 2020).

Three performance criteria are used to compare the performance of these methods. A first criterion measures the maximum absolute error between the fluxes computed from one of the methods above and the Radau method:

$$E^m = max\left\{ \left|\hat{F}_i^m(k) - \hat{F}_i^{radau}(k)\right|, k = 1, \dots, p \quad i = 1, \dots, n \right\}$$ **Eq. 11**

where $\hat{F}_i^m(k)$ is the $i^{th}$ flux computed with method $m$ for time step $k$. $E^m$ is measured in the unit of the reservoir fluxes, hence $m^3 s^{-1}$ for $CR$ and $BCR$, and $mm\ day^{-1}$ for the two remaining equations.

The second criterion is the maximum mass balance error between the fluxes computed with one of the methods and Radau:

$$B^m = max\left\{ \left|\frac{\sum_{k=1}^p \hat{F}_i^m(k) - \hat{F}_i^{radau}(k)}{\sum_{k=1}^p \hat{F}_i^{radau}(k)}\right|, i = 1, \dots, n \right\} \times 100$$ **Eq. 12**

$B^m$ is dimensionless and reported in percent. Finally, the third criterion compares the runtime of a method against the Radau runtime:

$$R^m = \frac{T^m}{T^{radau}} \times 100$$ **Eq. 13**

The runtime was measured on a laptop computer using a quadcore processor with a clock speed of 3GHz. Note that the runtime of QuaSoARe is assessed using a version of the code written in pure Python so that it can be compared with Radau and RK45 methods. A faster version of the code written in C available in the same package is recommended for application

purposes (Lerat, 2024).

Measuring runtime is dependent on the machine used and may thus lack generality. An alternative metric to measure computational speed was tested (not reported here) where runtime is assessed by the number of flux function evaluations following Clark and Kavetski (2010). This metric provided similar results compared to $R^m$ but is harder to interpret because QuaSoARe and the two discrete methods do not evaluate the same functions (analytical solutions for QuaSoARe and

original flux functions for RK45 and Radau). Consequently, the simple runtime metric $R^m$ is preferred here.



**Table 2: Characteristic of the case study catchments**

| Name | Outlet station | Upstream station | Hydraulic distance upstream-outlet [$km$] | 2022 Rainfall [$mm\ y^{-1}$] | 2022 PET* [$mm\ y^{-1}$] | 2022 Outlet peak [$m^3\ s^{-1}$] |
|---|---|---|---|---|---|---|
| Casino | Richmond River at Casino Site ID 203004, 1,790 $km^2$ | Eden Creek at Doubtful Site ID 203034, 555 $km^2$ | 32 | 1941 | 1455 | 2077 |
| Wiangaree | Richmond River at Wiangaree Site ID 203005, 702 $km^2$ | Richmond River at Lavelles Road Site ID 203056, 337 $km^2$ | 16 | 1908 | 1430 | 1486 |
| Eltham | Wilsons River at Eltham Site ID 203014, 223 $km^2$ | Byron Creek at Binna Burra Site ID 203012, 36 $km^2$ | 20 | 3291 | 1472 | 573 |
| Ewing Bridge | Coopers Creek at Ewing Bridge Site ID 203024, 148 $km^2$ | Coopers Creek at Repentance Site ID 203002, 52 $km^2$ | 22 | 3401 | 1440 | 770 |
| Fairmeadow | Coopers Creek at Fairmeadow Site ID 203060, 177 $km^2$ | Coopers Creek at Repentance Site ID 203002, 57 $km^2$ | 26 | 3360 | 1441 | 365 |
| Kyogle | Richmond River at Kyogle Site ID 203900, 899 $km^2$ | Richmond River at Wiangaree Site ID 203005, 710 $km^2$ | 19 | 1992 | 1435 | 1358 |

*Potential Evapotranspiration

## 5.3   Case study area

QuaSoARe performance is evaluated for 6 sites located in the Richmond River catchment in Eastern Australia, close to the city of Lismore. The catchments are presented in Table 1 and their location is shown in Figure 3. This area was chosen because it experienced a devastating flood in February 2022 which prompted in-depth analysis of the event (Lerat et al.,

2022). In addition, the maximum rainfall totals observed during this flood exceeded 700 mm in 24 hours. Such extreme values constitute a challenging test for ODE solvers as pointed out by La Follette et al. (2021). For each of catchment, the outlet station and one station located upstream of the outlet shown in Figure 3 are selected. The hourly streamflow data from the upstream station from the 1st February 2022 to 10th April 2022 are used to run the two routing models (CR and BCR). The daily catchment average climate data from the 1st January 2010 to 31st December 2022 are used to run the two

hydrological models (GR and GRM).

As highlighted in section 5.2, our aim is not to simulate hydrological processes accurately in these catchments, but only to obtain realistic forcing data for our numerical experiments.







**Figure 3: Location of the test catchments**

**6    Results**

**6.1    Simulation of the GR4J production store**

As an example of the application of QuaSoARe, this section details the results obtained with the GR4J production store (GR) presented in Table 1 for the Coopers Creek at Ewin Bridge catchment. The GR simulation is run with a storage scaling factor of 500 mm which is a common value for this parameter in Australia. Figure 4 shows the interpolation performed by





QuaSoARe to approximate the three fluxes of the GR model using 10 nodes. The plots show the storage level $S$ divided by the scaling factor $\theta$ in x-axis and the instantaneous flux in y-axis. The values for $P$ and $E$ required to compute the flux functions in Table 1 are set to 4.2 and 3.1 $mm\ day^{-1}$, respectively, which correspond to the mean daily rainfall and potential evapotranspiration over the simulation period. The interpolation error is shown as a thin black line on the same plot using the same x-axis and a secondary y-axis of different scale. In the three plots, the QuaSoARe flux appears to be

indistinguishable from the "true" flux, which is visually satisfying but not sufficient to guarantee an accurate solution as already seen in Figure 1. The interpolation error for the first two fluxes is smaller in magnitude than $2 \times 10^{-15}\ mm\ day^{-1}$ which is comparable to our machine precision. This is expected as the first two GR flux functions are quadratic function that can be interpolated exactly by QuaSoARe. Note that using ten nodes is superfluous in this case as two nodes would suffice. The third flux is a power function with an exponent of 5 which cannot be interpolated exactly. Consequently, the

interpolation error is larger, reaching a magnitude up to $2.5 \times 10^{-3}\ mm\ day^{-1}$. This error remains small and shows an oscillating behaviour that is characteristic of polynomial interpolation using nodes with constant spacing. A potential improvement on this point is discussed in Section 7.

Figure 5 shows the daily storage levels and fluxes from the GR reservoir for the year 2022 using the Radau (orange lines) method, considered as the "truth", and QuaSoARe method (blue lines) with ten interpolation nodes. In this figure, the Radau

and QuaSoARe simulations are visually indistinguishable which suggests that the interpolation errors shown in Figure 4 remain small enough to have a lasting impact. The simulation errors are shown as thin black lines using a secondary y-axis and confirm that, despite the low number of nodes used, QuaSoARe can simulate the dynamic of the GR store and all its fluxes with errors lower in magnitude to $5 \times 10^{-3}\ mm\ day^{-1}$, which is negligible compared to typical errors of climate input data which are rarely below $10^{-2}\ mm\ day^{-1}$ in a research catchment and probably much higher in our study

catchment. All flux errors exhibit oscillating patterns already noted in the interpolation error shown in Figure 4.c, which are likely due to a combination of oscillating interpolation errors and the alternance of wet and dry days leading to sudden variations in actual evapotranspiration visible in Figure 5.c. This suggests that quantifying simulation errors in a highly non-linear reservoir equation is not straightforward and depends on both the quality of the numerical solution and statistical characteristics of input forcings. More generally, the GR flux functions constitute a challenging case study because they

combine the slow dynamic of the storage visible in Figure 5.a and rapid changes of actual ET seen in Figure 5.c. QuaSoARe appears to resolve both with a high accuracy.





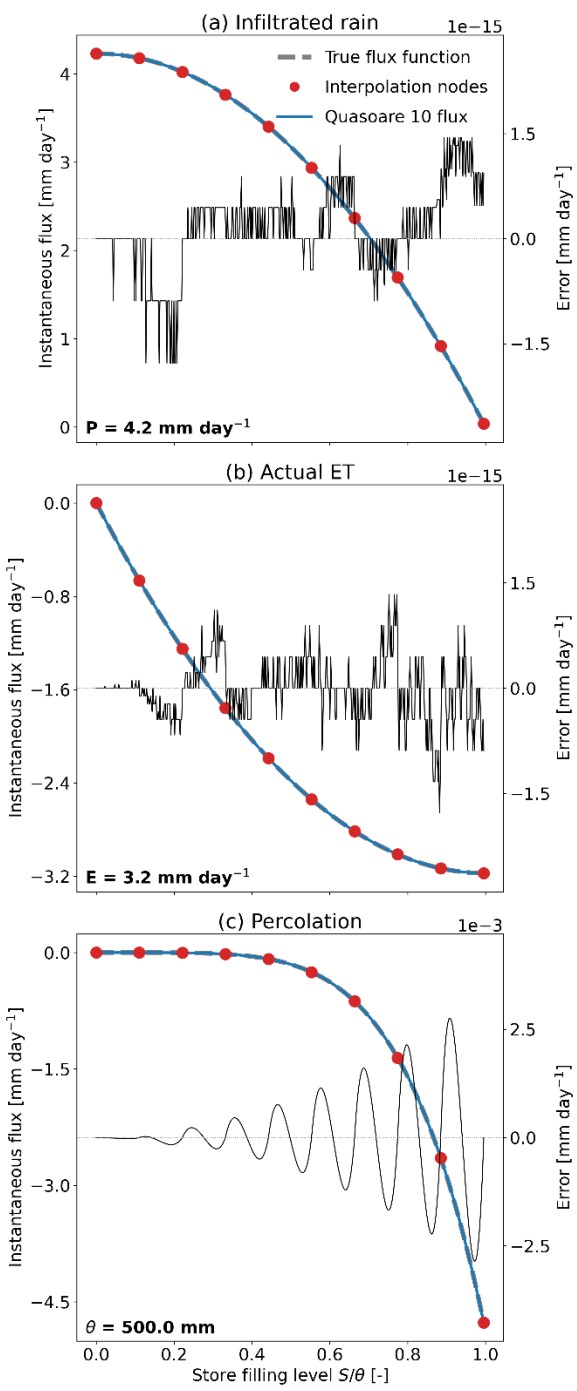

**Figure 4: True and approximated flux functions of the GR reservoir equation with a scaling factor $\theta$ set to 500 mm. $P$ and $E$ variables are set to 4.2 and 3.1 mm and QuaSoARe interpolation is using 10 nodes.**





**Figure 5: Radau and QuaSoARe storage level (plot a) and three fluxes (plots b to c) for the GR reservoir with a scaling factor $\theta$ set to 500 mm and using data from the Coopers Creek at Ewing Bridge catchment. QuaSoARe is configured with 10 interpolation nodes.**







**Figure 6: Comparison of performance metrics between the RK45 method and QuaSoARe using 10, 50 and 500 interpolation nodes. The Radau method is used as a reference simulation against which performance metrics are computed.**

## 6.2 Performance assessment

Expanding the analysis of the previous section, Figure 6 presents statistics of the three performance metrics introduced in Section 5.2. The metrics are computed for the four models (CR, BCR, GR and GRM, see Table 1), the six study catchments (see Table 2), 10 values of the storage capacity parameter $\theta$ for each model and five ODE solvers including the Radau and RK45 methods, and three configurations of QuaSoARe using 10, 50 and 500 nodes. Overall, 60 simulations are produced for



each combination of model and ODE solver. The Radau method is considered as the reference against which the error of other methods is computed. The numbers shown in each plot of Figure 6 report the median value of the metric over the 60

corresponding simulations while the blue boxes show their min to max range.

All the plots in Figure 6 reveal that QuaSoARe configured with a high number of nodes (500) lead to simulations that are extremely close to the Radau outputs. The median absolute simulation error of QuaSoARe ($E^m$) varies between $4.4 \times 10^{-6} \; mm \; day^{-1}$ for the GR model (Figure 6.g) to $9.4 \times 10^{-5} m^3 s^{-1}$ for the BCR model. Both values are several orders of magnitude lower than observation errors (e.g. Chubb et al. (2016) report root mean squared errors of gridded daily rainfall

data above $4.5 \; mm \; day^{-1}$), which means that simulations from QuaSoARe and Radau can be considered interchangeable from a modeller's perspective. Importantly, these results also hold for the mass balance metric $B^m$ where the average difference between the flux totals from QuaSoARe and Radau remains lower than $2 \times 10^{-6}$ % (Figure 6.k). At the same time, the QuaSoARe runtime is only a fraction of the Radau runtime with median values varying from 3.8% for the CR and BCR models (Figure 6.c and f) to 11.8% for GRM (Figure 6.l). Finally, QuaSoARe performance are better than RK45 for

most model and performance metrics. For example, the maximum flux error of QuaSoARe is several orders of magnitude lower than RK45 except for the CR model where both reaches error close to $1.5 \times 10^{-5} \; m^3 s^{-1}$ (Figure 6.a).

When a lower number of nodes is used, the errors of QuaSoARe increase and worsen the performance metrics. For example, the median maximum error for the GR model shown in Figure 6.g increases from $4.4 \times 10^{-6} m^3 s^{-1}$ when using 500 nodes to $3.1 \times 10^{-3} \; m^3 s^{-1}$ with 10 nodes. This degradation is expected as the interpolation of the reservoir functions worsens

with fewer nodes. Nonetheless, the errors remain insignificant compared to observation errors which suggests that a low number of nodes remains an attractive configuration when applying QuaSoARe to hydrological models. The runtime efficacy of QuaSoARe when using ten nodes is noticeable with values remaining lower than 3% of the Radau runtime for the four models.

We highlight that the results presented here correspond to challenging test cases with highly non-linear flux functions. For

example, the GRM reservoir includes functions with polynomial of order up to seven. In addition, the selected catchments exhibit particularly difficult hydro-climate regime to simulate with maximum rainfall intensity exceeding several hundred millimetres per day.

## 7  Discussion

The QuaSoARe method presented in this paper constitutes a valuable alternative to existing numerical schemes when solving

the scalar reservoir equation for three reasons: first the algorithm is simple to understand and code, relying only on the interpolation of the reservoir functions by piecewise quadratic polynomials. Consequently, we believe that it could be integrated in existing modelling platforms and help strengthen the numerical solution of hydrological models without introducing a significant burden in terms of code maintenance. Second, the method is much faster than standard alternatives such as the RK45 (explicit) and Radau (implicit) schemes tested in this paper. QuaSoARe reduces the runtime by a factor of



20 to 50 compared to Radau depending on the model. This point further reinforces the value of the method for hydrological modelling tasks requiring repeated model evaluation such as automated calibration or data assimilation. Finally, the configuration of QuaSoARe can be modified simply by changing the number of interpolation nodes to vary between highly accurate but slow simulations (say with 500 nodes) to lower accuracy and fast runtime (say with 10 nodes). As a result, the modeller remains in control of the algorithm with a single configuration parameter which has a simple interpretation.

The QuaSoARe performance reported in this paper are satisfactory but one may wonder if there could be ways to improve them further without changing the core of the algorithm. The first point that could be improved is the quality of the interpolation. Our method relies on piecewise quadratic polynomials which could be replaced by other, potentially more flexible functions. However, the requirements of QuaSoARe in terms of interpolation functions are stringent because they should be at the same time stable by linear combination (to compute all fluxes) and leading to an analytical solution of the

approximate reservoir equation. An interesting alternative to quadratic polynomials which was explored in early version of this work is a combination of exponential functions. This choice was later abandoned because it was prone to numerical overflow and did not provide significant performance improvements. Nonetheless, we believe that other functions could meet the QuaSoARe requirements and help improve its performance in difficult situations, for example when the flux functions are rational fractions which are notoriously difficult to interpolate with polynomials (Berrut and Trefethen, 2004).

Related to this point, one may wonder about the performance of a linear interpolation in comparison to its quadratic counterpart shown in this paper. Tests not shown here revealed that the linear interpolation worsens the performance by an approximate factor of ten with a similar runtime. Interestingly, the difference between the two seems to persist even when using a large number of nodes, hence when the interpolation of both linear and quadratic is expected to be close. Overall, we recommend the quadratic interpolation, but kept the linear approach as a degraded functionality in our code (Lerat, 2024).

Several ideas could be explored to improve QuaSoARe further. In this paper, the interpolation nodes are placed at equal intervals between the two extremes $\alpha_1$ and $\alpha_m$. This could be easily modified to find the nodes providing the lowest interpolation errors. This might reduce the oscillations of the interpolation errors visible in Figure 4.c. However, this would also increase the complexity of the algorithm, and we favoured simplicity in this first version of QuaSoARe. Finally, we flag that our method could be combined with a discrete method similarly to the exponential integrator approach of Pope (1963).

This is likely to improve performance further but would also require the use of a discrete method which we discarded in our introduction. However, this could constitute an interesting avenue for a modeller who is already familiar with discrete methods.

Beyond potential improvements of QuaSoARe, we highlight that the flux functions defined in an empirical modelling context like most rainfall-runoff models remains arbitrary and are not derived from physically-based equations. In this case,

one could argue that piecewise quadratic functions could themselves constitute valid flux functions. This approach would have two merits: first the instantaneous model equations could be characterised by a small set of coefficients (the interpolation coefficients), hence becoming fully parametric and amenable to optimisation or data assimilation. Second, the model could be solved exactly with QuaSoARe. As a result, models formulated in this way could be integrated in flexible



modelling environments. This avenue is currently explored to extend the work of Lerat et al. (2024) where improvement of
model structure is obtained by correcting state equations via data assimilation methods.

## 8    Conclusion

This paper presents a simple numerical algorithm called QuaSoARe to solve the instantaneous reservoir equation and
provide a basis to build hydrological models that can integrate this equation and generate fast and accurate solutions. The
method was tested on a range of highly nonlinear models that are representative of hydrological models in use. Its
performance suggests that the method matches the accuracy of a high-order implicit discrete method while requiring a
fraction of its runtime. Yet, the method algorithm is simple and can be described with a short pseudo-code included in our
paper.

The method is limited by its applicability to scalar equations and by the quality of quadratic interpolation which underlies the
method analytical solution. To model a series of reservoir, we suggest implementing sub-time step integration. Higher
accuracy in the interpolation of reservoir functions can always be achieved by increasing the number of interpolation nodes.
Our results showed that the use of 10 to 50 nodes leads to an accuracy level that is order of magnitude smaller than typical
error in observation data.

## Competing interests

The contact author declares no competing interests.

**Acknowledgements**

This study was funded by Rous County Council as part of the Richmond and Wilsons Rivers Flood Risk Mitigation Project.
The contribution from Tewfik Sari and Vazken Andreassian from INRAE, who pointed out references to the TPLA method
and commented on early versions of this manuscript is greatly acknowledged. Comments provided by Justin Hughes,
CSIRO, and Charles Perrin, INRAE, and site map prepared by Steve Marvanek, CSIRO, are also acknowledged.

## Code and data availability

The QuaSoARe method is released as part of a python package written by Lerat (2024). The package contains two
implementations of the method written in the Python and C languages. The C version is recommended due to its significantly
faster runtime. All scripts and hydro-climate data used to generate the results in this paper are included in the code
repository.



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





## Appendix A: Interpolation coefficients

For each function $f_i$ and each interval $[\alpha_j, \alpha_{j+1}]$, we seek values of the coefficients $a_{i,j}$, $b_{i,j}$ and $c_{i,j}$ so that the approximated function $f_i^*$ matches $f_i$ as best as possible. This is achieved by matching $f_i$ at the two points $\alpha_j$ and $\alpha_{j+1}$ and at their mid-point. Assuming that $\alpha_j < \alpha_{j+1}$, quadratic interpolation leads to

$$a_{i,j} = \frac{2f_0 + 2f_1 - 4f_m}{\left(\alpha_{j+1} - \alpha_j\right)^2} \qquad \text{Eq. 14}$$

$$b_{i,j} = -2\alpha_j\, a_{i,j} + \frac{4f_m - 3f_0 - f_1}{\alpha_{j+1} - \alpha_j} \qquad \text{Eq. 15}$$

$$c_{i,j} = \alpha_j^2\, a_{i,j} - \alpha_j\, \frac{4f_m - 3f_0 - f_1}{\alpha_{j+1} - \alpha_j} + f_0 \qquad \text{Eq. 16}$$

Where

$$f_0 = f_i(\alpha_j), \qquad f_1 = f_i(\alpha_{j+1}), \qquad f_m = f_i\left(\frac{\alpha_j + \alpha_{j+1}}{2}\right) \qquad \text{Eq. 17}$$

This solution is straightforward but can lead to a function $f_i^*$ that is non-monotonous, e.g. when interpolating a function $f_i$ that presents sharp transitions or asymptotes. This situation can create an issue related to the steady state solution of Eq. 1 as discussed in Section 4. To avoid this undesirable effect, we replace the value of $f_m$ used in Eq. 14 to Eq. 16 by a value $f_m^*$ defined as follows:

$$f_m^* = \max\left(f_m^-, \min(f_m^+, f_m)\right) \qquad \text{Eq. 18}$$

Where the two bounds $f_m^-$ and $f_m^+$ are given by:

$$f_m^- = \min\left(\frac{3f_0 + f_1}{4}, \frac{f_0 + 3f_1}{4}\right) \qquad \text{Eq. 19}$$

$$f_m^+ = \max\left(\frac{3f_0 + f_1}{4}, \frac{f_0 + 3f_1}{4}\right) \qquad \text{Eq. 20}$$

It can be verified that using $f_m^*$ instead of $f_m$ in Eq. 14 to Eq. 16 leads to a monotonous quadratic function $f_i^*$.

## Appendix B: Analytical solution of the approximated reservoir equation

In this appendix, we solve the approximate reservoir equation (Eq. 9) over an interval $[t_0, t_1]$ such that $\forall t, \ S(t) \in [\alpha_j, \alpha_{j+1}]$ starting from the initial condition $S(t_0) = S_0$. This solution is obtained by separating the variables leading to the following integral equation

$$\int_{S_0}^{S_1} \frac{dS}{A_j\, S^2 + b_{\circ,j}\, S + c_{\circ,j}} = t \qquad \text{Eq. 21}$$





Where $S_1 = S(t_1)$. Here we assume that either $A_j \neq 0$ or $B_j \neq 0$ otherwise the interpolation function $f_i^*$ is simply the constant $C_j$ leading to

$$S(t_1) = S_0 + C_j\,\tau \qquad \text{Eq. 22}$$

Where $\tau = t_1 - t_0$. For other configurations, the solutions of Eq. 21 can be obtained from any calculus textbook. They are presented succinctly in Table 3. In this table, the subscripts "$j$" is dropped to simplify notations. Function $\nu$ mentioned in the second column of Table 3 is defined as the following primitive:

$$\nu\big(A_j, B_j, C_j, s\big) = \int \frac{dS}{A_j\,S^2 + B_j\,S + C_j} \qquad \text{Eq. 23}$$

This function is used in our algorithm to compute the times $t_l$ and $t_u$ mentioned in Cases 2 and 3 in Section 2.3. Finally, Table 3 uses the following expressions to simplify notations further:

$$\bar{S} = -\frac{B_j}{2A_j} \qquad \text{Eq. 24}$$

$$\Delta = B_j^2 - 4A_j C_j \quad \sigma_\Delta = sgn(\Delta) \quad q_\Delta = \frac{\sqrt{|\Delta|}}{2} \qquad \text{Eq. 25}$$

$$\omega_\Delta(x) = \begin{cases} \tan(x) & if\ \Delta < 0 \\ \tanh(x) & if\ \Delta > 0 \end{cases} \qquad \text{Eq. 26}$$

$$\eta_\Delta(x) = \begin{cases} atan(x) & if\ \Delta < 0 \\ -atanh(x) & if\ \Delta > 0\ and\ |x| < 1 \\ -atanh(1/x) & if\ \Delta > 0\ and\ |x| > 1 \end{cases} \qquad \text{Eq. 27}$$

Where $sgn$ is the sign function.

We now focus on finding an expression for the total of the $i^{th}$ flux introduced in Eq. 2. Combining the solution of Eq. 21 given in the third column of Table 3, noted $S(t)$, with Eq. 2 gives

$$O_i^*(t_0, t_1) = \int_{t_0}^{t_1} f_i^*(S(t))\,dt = a_{i,j}\int_{t_0}^{t_1} S(t)^2\,dt + b_{i,j}\int_{t_0}^{t_1} S(t)\,dt + c_{i,j}\tau \qquad \text{Eq. 28}$$

Where $\tau = t_1 - t_0$. Note that in the above equations, the integral boundaries used in Eq. 2 ($[0, \delta]$) are generalized to $[t_0, t_1]$ because fluxes are often computed on an smaller interval than $[0, \delta]$ in the algorithm described in Section 2.3. Solving Eq. 28 is equivalent to integrating $S(t)$ and $S(t)^2$ between $t_0$ and $t_1$. To compute these two integrals, we consider different cases for $A_j$ and $B_j$ starting with $A_j = B_j = 0$. In this case, we obtain from Eq. 22:

$$\int_{t_0}^{t_1} S(t)\,dt = S_0\tau + C_j\frac{\tau^2}{2} \qquad \text{Eq. 29}$$

$$\int_{t_0}^{t_1} S(t)^2\,dt = S_0^2\tau + S_0 C_j\tau^2 + C_j^2\frac{\tau^3}{3} \qquad \text{Eq. 30}$$

We now assume that $A_j = 0$ and $B_j \neq 0$. The two integrals can be obtained by manipulating and integrating Eq. 9 as follows:



$$\int_{t_0}^{t_1} \frac{dS}{dt}\, dt = B_j \int_{t_0}^{t_1} S(t)\, dt + C_j \tau \;\Rightarrow\; \int_{t_0}^{t_1} S(t)\, dt = \frac{1}{B_j}(S_1 - S_0 - C_j \tau)$$

Eq. 31

$$\int_{t_1}^{t_2} S(t)\frac{dS}{dt}\, dt = B_j \int_{t_1}^{t_2} S(t)^2\, dt + C_j \int_{t_1}^{t_2} S(t)\, dt$$

Eq. 32

$$\Rightarrow\; \int_{t_0}^{t_1} S(t)^2\, dt = \frac{1}{B_j}\left(\frac{S_1^2 - S_0^2}{2} - C_j \int_{t_0}^{t_1} S(t)\, dt\right)$$

With these two equations, integrating $S$ is done first using Eq. 31. This integral is then used in Eq. 32 to compute the integral of $S^2$.

Finally, we now address the general case where $A_j \neq 0$. Following a similar approach than above, we re-arrange and integrate Eq. 9 as follows:

$$\int_{t_0}^{t_1} \frac{dS}{dt}\, dt = A_j \int_{t_0}^{t_1} S(t)^2\, dt + B_j \int_{t_0}^{t_1} S(t)\, dt + C_j \tau$$

Eq. 33

$$\Rightarrow\; \int_{t_0}^{t_1} S(t)^2\, dt = \frac{1}{A_j}\left(S_1 - S_0 - B_j \int_{t_0}^{t_1} S(t)\, dt - C_j \tau\right)$$

Consequently, the integration of $S^2$ can be deducted from the one of $S$. For this last integral, we use the expression of $S$ given in the third column of Table 3. If $\Delta = 0$ (see row before last in Table 3), we obtain:

$$\int_{t_0}^{t_1} S(t)\, dt = \bar{S}\, \tau - \frac{1}{a}\log[1 - a\, \tau\, (S_0 - \bar{S})]$$

Eq. 34

A value $\Delta \neq 0$ leads to (see last row in Table 3)

$$\int_{t_0}^{t_1} S(t)\, dt = \bar{S}\, \tau + \frac{1}{2a}\log\big(1 - \sigma_\Delta\, \omega_\Delta^2(q_\Delta \tau)\big) - \frac{1}{a}\log\left(1 - a\frac{S_0 - \bar{S}}{q_\Delta}\, \omega_\Delta(q_\Delta \tau)\right)$$

Eq. 35





645

**Table 3: Solutions of approximate reservoir equation**

| Case | Primitive $v(A, B, C, s)$ | Solution $S(A, B, C, S_0, t)$ | Domain of validity of solution $S(A, B, C, S_0, t)$ noted $[0, T_m[$ |
|---|---|---|---|
| $A = 0$ $B \neq 0$ | $\dfrac{1}{B}\log\lvert B\,s + C\rvert$ | $-\dfrac{C}{B} + \left(S_0 + \dfrac{C}{B}\right)\exp(b\,t)$ | $T_m = +\infty$ |
| $A \neq 0$ $\Delta = 0$ | $-\dfrac{1}{A(s - \bar{S})}$ | $\bar{S} + \dfrac{S_0 - \bar{S}}{1 - A\,t\,(S_0 - \bar{S})}$ | $T_m = +\infty$ if $A(S_0 - \bar{S}) \leq 0$ <br> Otherwise <br> $T_m = \dfrac{1}{A(S_0 - \bar{S})}$ |
| $A \neq 0$ $\Delta \neq 0$ | $\dfrac{1}{q_\Delta}\eta_\Delta\left(A\dfrac{s - \bar{S}}{q_\Delta}\right)$ | $\bar{S} + \dfrac{S_0 - \bar{S} - \sigma_\Delta \dfrac{q_\Delta}{A}\,\omega_\Delta(q_\Delta t)}{1 - A\dfrac{S_0 - \bar{S}}{q_\Delta}\,\omega_\Delta(q_\Delta t)}$ | If $\Delta < 0$ <br> $T_m = \dfrac{1}{q_\Delta}\left[\dfrac{\pi}{2} - \eta_\Delta\left(A\dfrac{S_0 - \bar{S}}{q_\Delta}\right)\right]$ <br> Otherwise, <br> $T_m = \begin{cases} +\infty & if\ A(S_0 - \bar{S}) < q_\Delta \\ -\dfrac{1}{q_\Delta}\eta_\Delta\left(A\dfrac{S_0 - \bar{S}}{q_\Delta}\right) & otherwise \end{cases}$ |