# Peer review of "Technical note: Quadratic solution of the approximate reservoir equation (QuaSoARe)"

_EGUsphere, 2024_

## Referee Comment (RC2)

**Comment on egusphere-2024-3184**

by J. Lerat

January 8, 2025

**1 General comments**

The technical note proposed by J. Lerat introduces a new method for the integration of a scalar Ordinary Differential Equation (ODE), describing the evolution of a conceptual component such as a reservoir in a lumped hydrological model. The rationale for the development of the method is that, given this ODE formulated in state-space, it is difficult to compute together both the evolution of the single state variable (denoted $S$) over a timestep $[t, t+\delta]$, as well as the different outputs from the system *integrated over the timestep $\delta$*. The QuaSoARe method uses analytical (quadratic) substitutes for the instantaneous fluxes entering/leaving the reservoir.

The paper is written in a very straightforward way and is relatively easy to follow with some familiarity in analytical and numerical integration (apart from the very first part where continuity assumptions are stated, see comments below). I am overall supportive of the approach which has a great potential for speeding up current formulations of many lumped models, but I still have questions regarding the kind of solutions which should be put forward to cure the problems raised by Clark and Kavetski about the numerical soundness of many such models, in the case of multistate model structures.

**2 Specific comments**

**2.1 Discrete time vs. state-space formulations and the problem of sub-timestep distribution of inputs / outputs**

I think that the introduction somehow lacks a brief presentation of the context in which unbalanced, cumulative outputs from model reservoirs could happen. Equation 1

$$\dot{S} = \frac{dS}{dt} = \sum_i f_i(S, \tilde{V})$$

is straightforward to understand but it already constitutes a "zoom" on a single model element (reservoir), along with two assumptions:

(i) the right-hand-side of this state-space formulated equation does not depend on any other state (e.g. the storage $R$ of an other reservoir),

(ii) in the absence of information about the sub-timestep distribution of forcing variables such as $\tilde{V}$, they are assumed constant over the timestep so that the equation does not explicitly depend on the independent time variable $t$: we can write $f_i(S, \tilde{V} = \text{cst})$ rather than $f_i(S(t), t)$. Restricting the analysis to such autonomous ODEs (Clark and Kavetski, 2010) is very common and is a lesser loss of generality than assumption (i), but it is still worth mentionning.

Assumption (i) can basically be met with operator splitting (OS) procedures, and this point will be discussed later. My first question would be: in validating model outputs against observations (e.g. runoff, actual evapotranspiration, recharge, etc.), why wouln't Eq. 3 be suitable for *flux* comparison, and why do we absolutely need to compute integrated quantities such as the $O_i's$ (runoff *depth* over $\delta$, evapotranspiration *depth*, recharge *depth*, etc.) which have dimension L (length), rather than to their instantaneous counterparts $\frac{dO_i}{dt}$ (runoff *rate*, evapotranspiration *rate*, etc. with dimension $L \cdot T^{-1}$) at successive times $t, t + \delta, t + 2\delta$, etc.

My understanding is that models targeted by the QuaSoARe procedure are essentially *balance* models whose purpose is mainly to ensure mass conservation at quite large time steps (e.g., daily), i.e. to yield a value for $S[\delta]$ and a set of values for the $\{O_i\}_{1 \leq i \leq n}$ such that:

$$(B) \begin{cases} S(\delta) - S(0) \approx \sum_{i=1}^{n} \int_0^\delta f_i(S, \tilde{V}) dt \quad \text{(approximately)} \\[3em] S(\delta) - S(0) = \sum_{i=1}^{n} O_i \quad \text{(strict equality)} \end{cases}$$

QuaSoARe clearly meets these requirements, because it uses an analytical substitute $\hat{f}_i(S, \tilde{V})$ for each flux, such that

$$(\hat{B}) \begin{cases} S(\delta) - S(0) = \sum_{i=1}^{n} \int_0^\delta \hat{f}_i(S, \tilde{V}) dt \approx \sum_{i=1}^{n} \int_0^\delta f_i(S, \tilde{V}) dt \quad \text{(approximately)} \\[2em] O_i = \int_0^\delta \hat{f}_i(S, \tilde{V}) dt \quad \forall i \\[2em] S(\delta) - S(0) = \sum_{i=1}^{n} \int_0^\delta \hat{f}_i(S, \tilde{V}) dt = \sum_i O_i \quad \text{(strict equality yielded by previous eqns)} \end{cases}$$

Clearly, solving flux equations individually is a *sufficient* condition to fullfill the conditions (B), but I rather disagree on the fact that it is a *necessary* one, as stated in lines 60–70 ("*Finally, jointly solving Eq. 1 and Eq. 2 using a numerical solver* ***requires*** *transforming the scalar equation Eq. 1 to a system of differential equations by adding one scalar equation for each flux*"). If we have a procedure to solve the single, state-space formulated ODE in $S$, attributing residual errors in cumulative balance to any of the outputs would yield the same result at lower computational cost. If we chose an output $O_{i_0}$ for balance error compensation, there are several possible fixes allowing to fullfill conditions (B), such as:

$$(B') \begin{cases} O_i &= f_i\left(\dfrac{S(0)+S(\delta)}{2}, \tilde{V}\right) \quad \forall i \neq i_0 \\[2ex] O_{i_0} &= S(\delta) - S(0) - \displaystyle\sum_{i \neq i_0} O_i \end{cases} \qquad (B'') \begin{cases} O_i &= \dfrac{1}{2}\left[f_i\left(S(0),\tilde{V}\right) + f_i\left(S(\delta),\tilde{V}\right)\right] \quad \forall i \neq i_0 \\[2ex] O_{i_0} &= S(\delta) - S(0) - \displaystyle\sum_{i \neq i_0} O_i \end{cases}$$

Using the language of atmospheric modeling we might say that the $O_i$'s are some kind of *diagnostic* variables that could be computed at the end of the simulated time period, $S$ being the only variable truly belonging to the category of *state variables*. Adding this distinction in the formulation of the problem could improve the paper and help modelers think about the way their models are written (either discrete time, or state-space). We must acknowledge that this issue about *individual fluxes* computations raised by J. Lerat is even less discussed in the hydrological litterature than that of numerical scheme adequacy. The issue of solution constraints (storage remaining non-negative, as well as fluxes such as evaporation, etc.) is discussed in Clark and Kavetski (2010) along with the semi-implicit Euler scheme applied to an autonomous system of $N_s$ coupled ODEs in the form:

$$\dot{\mathbf{S}} = \mathbf{g}(\mathbf{S})$$

These authors propose to compute the $k$-th *individual flow volume* for store $i$ using the end-of-time-step (index $n+1$) value of the instantaneous flux multiplied by the time step duration, i.e.

$$O_i^{(k)} = \Delta t \; g_i^{(k)}(\mathbf{S}^{n+1})$$

They acknowledge that "ad hoc fixes such as zeroing negative fluxes works in simple cases but are not a satisfactory general solution", and again the issue of multi-state models arises since "in multistate models where the states are couple via the fluxes, fixing the violation in one state can impact on the feasibility of another". This would be my main question about a possible extension of QuaSoARe to the case of multistate models: can we avoid operator splitting, and if no, shouldn't we still favor direct integration of the full, coupled system of ODEs describing the evolution of the state *vector* $\mathbf{S} \in \mathbb{R}^m$ using the semi-implicit Euler (SIE) scheme for example?

**2.2   Mathematical assumptions on the flux functions $f_i(S, \tilde{V})$**

The introduction of the paper quickly gets us into the swing of things with the mathematical assumptions stated p.2 (l. 36–41). It is a bit difficult to understand the practical implications of the Lipschitz-continuity assumption: does it mean that some functions are not admissible as flux functions? If we consider the classical equation of a draining tank, according to Bernoulli equation the spout exit velocity is given by:

$$u(t) = \sqrt{2gh(t)}$$

where $h(t)$ is the water depth in the reservoir. Reformulating this as a storage-output equation using $Q = a \cdot u$ with $a$ the spout exit section and using $A$ the base section of the reservoir such that $h(t) = S(t)/A$, we have

$$Q(t) = \frac{dS}{dt} = a\sqrt{2g\frac{S(t)}{A}}$$

This function is not Lipschitz-continuous on the possible range of storage values since the derivative of $\sqrt{x}$ is unbounded as $x$ tends to zero; does it mean that we have to check each flux function?

---

## Author Comment (AC1)

**Technical note: Quadratic solution of the approximate reservoir equation (QuaSoARe) - Supplement**

Julien Lerat [1]

[1]CSIRO Environment, Canberra, ACT, 2601, Australia

5   *Correspondence to*: Julien Lerat (julien.lerat@csiro.au)

**S1.  Application of QuaSoARe to flux functions that are not Lipschitz continuous**

QuaSoARe is a method designed to solve the reservoir equation given an initial condition and a set of flux functions. The method is developed based on the assumption that the flux functions are all Lipschitz continuous (see Section 1 of the paper) which restricts the choice of potential flux functions.  This section explores the impact of applying QuaSoARe to flux

10   functions that are not meeting this condition. More precisely, a variant of the routing models CR and BCR presented in Table 1 of the paper is considered where the reservoir equation is:

$$\frac{dS}{dt} = Q_{inflow} - Q_{ref}\left(\frac{S}{\theta}\right)^{\kappa} \qquad \textbf{Eq. 1}$$

Where $Q_{inflow}$ and $Q_{ref}$ are the river reach inflow and reference flow ($m^3\ s^{-1}$), respectively, $\theta$ is the scaling factor set to 43 200 $Q_{ref}$ and $\kappa$ is an exponent set to 0.5. The catchment selected here is Coopers Creek at Ewin Bridge (203024). In this case, the second flux function in the right-hand side of Eq 1 is not Lipschitz continuous in $S = 0$ because it becomes

15   infinitely steep at this point due to the fact that $\kappa$ is strictly lower than 1.

[Figure]

**Figure S1: True and approximated flux functions of the routing reservoir equation with QuaSoARe interpolation using 3, 10 and 50 nodes.**

Applying QuaSoARe requires to first interpolate the flux functions using a given set of interpolation nodes. Figure S1 shows

20   the result of this process when using 3 (Figure S1.a), 10 (Figure S1.b) and 50 (Figure S1.c) nodes. Figure S1.a highlights the difficulty of interpolating a non-Lipschitz continuous function using a few quadratic polynomials: large discrepancies between the true (grey) and approximated (blue) functions appear close to the point $S = 0$. Increasing the number of nodes reduces these discrepancies significantly but cannot eliminate them completely as can be seen in Figure S1.c.

[Figure]

**Figure S2: Radau and QuaSoARe storage level (plot a) and outflow flux (plots b) for the routing reservoir using data from the Coopers Creek at Ewing Bridge catchment. QuaSoARe is configured with 3, 10 and 50 interpolation nodes.**

Once the interpolation is done, QuaSoARe can be run. Figure S2 shows the simulation corresponding to four methods of integration: the Radau scheme (see Section 5.2 of the paper), and QuaSoARe using 3, 10 and 50 nodes. This figure reveals that QuaSoARe simulation using 3 nodes (light blue line) introduces large errors in the simulated outflow compared to the Radau outputs (orange line). This is not surprising considering the discrepancies between the true and approximated fluxes shown in Figure S1. However, the QuaSoARe simulation using 50 nodes (dark blue line) remains relatively close the Radau simulation in both plots of Figure S2, which suggests that it is possible to obtain a reasonable simulation using QuaSoARe even if the flux functions are not Lipschitz continuous. This is, however, highly dependent on the case considered and will probably require a much higher number of interpolation nodes compared to reservoirs where flux functions are smoother.

**S2. Approximate computation of flux totals**

In section 1.2 of the paper, it is mentioned that the flux totals $O_i$ could be computed using simplified quadrature method as an alternative to expanding the reservoir equation into a system of differential equation or using QuaSoARe. If we assume that the reservoir equation is solved, i.e. that the two values $S_0$ (initial condition) and $S(\delta)$ are known, two of these quadrature methods suggested by one of the anonymous reviewers could be expressed as follows:

$$Mid\ point\ method: O_i = \int_0^\delta f_i(S, \tilde{V})\, dt \approx \delta\, f_i\left(\frac{S_0 + S(\delta)}{2}, \tilde{V}\right)$$

**Eq. 2**

$$Mid\ flux\ method: O_i \approx \delta\, \frac{f_i(S_0, \tilde{V}) + f_i(S(\delta), \tilde{V})}{2}$$

**Eq. 3**

Although computationally expeditive, both methods introduce an additional approximation to the solution of the reservoir equation which can lead to large errors if this approximation is poor. As an example, we applied the two approximate methods to GR4J production store (see Table 1 of the paper) when integrated with the Radau ODE solver (see Section 5.2 of

45      the paper) and compared them with the system expansion method indicated in the paper (see Eq. 3). Note that we did not use QuaSoARe in this example.

     The catchment selected is Coopers Creek at Ewin Bridge (203024). The store capacity $\theta$ is set to 50mm, which is outside of the parameter range reported in Table 1 of the paper (100 to 1000 mm). Based on our experience, such a small value of $\theta$ is rare in practice, but can happen during a calibration phase when a large number of parameters are tested. Intuitively, a GR4J

50      store with a small capacity receiving a large rainfall input will react quickly and show large variations of the flux functions over the time step.

[Figure]

**Figure S3: Comparison of flux computation using ODE system expansion method proposed in the paper and two approximate**
55      **quadrature methods.**

     Figure S3 shows the simulations of the store level (figure a) and the three fluxes: infiltrated rain (b), actual evapotranspiration (d), and percolation (f). The Radau fluxes (orange lines) are compared with fluxes computed with the two approximate methods: mid-point method shown (green lines) and mid-flux method (purple lines). Scatter plots of Radau versus approximated values are shown in figures c, e and g using the same colour scheme.

60      Overall, the approximated fluxes appear close to the Radau fluxes, especially for the actual ET flux where the three lines are visually indistinguishable. For infiltrated rain, the three flux computation methods remain very close except for large rainfall events, especially on the 24[th] February where the Radau flux is less than 20mm whereas both approximated methods lead to values exceeding 40mm. The scatter plot in figure (c) confirms the large errors introduced by the two approximate methods with point deviating significantly from the 1:1 line. To understand the reason for these discrepancies, the Radau integrator

65      was run at a finer time step of 30min during the 24[th] February starting from the initial condition extracted from the daily

simulation. The resulting half-hourly simulations of store level and infiltrated rain are shown in Figure S4. Note that a constant rainfall rate is used throughout the day to match with the daily simulation.

[Figure]

**Figure S4: GR4J simulations obtained with the Radau integrator for the 24th of February. Figure a shows the production store level. Figure b shows the infiltrated rain flux times series and its cumulative sum on a secondary y axis.**

Figure S4 reveals that the storage level $s(t)$ increases significantly during the simulation with most of the increase occurring during the first half of the day. As a result, the infiltrated rain, which is a decreasing quadratic function of $s$ (see Table 1 of the paper), decreases quickly at first and progressively more slowly to reach a value close to 0 at the end of the day. Both store and flux are clearly not following a linear trend, which explains why the two approximate flux computation methods severely overestimate the Radau flux for this day.

Overall, the approximate computation methods appear valid if the store and fluxes do not vary significantly during the time step of integration. However, it is difficult to guarantee that such sudden changes will not occur and, hence, degrade the simulation quality for certain flow regimes (particularly high flow regimes). For this reason, we recommend computing fluxes analytically, like in QuaSoARe, or using the ODE integrator in conjunction with Eq 3, as indicated in Section 1.2 of the paper.

---

## Author Response (AR1)

This document presents the response to the two reviewers of the paper "Quadratic solution of the approximate reservoir equation (QuaSoARe)"

The following changes were introduced in the paper following the reviewer comments:

- Fixed multiple typos and grammatical errors.
- Added several paragraphs to clarify the points raised by reviewer 2 related to
    (1) the type of model solved by QuaSoARe,
    (2) the implication of imposing Lipschitz continuity to flux functions,
    (3) the different nature of variable S (state variable) and O (diagnostic variable),
    (4) the possibility to estimate O using quadrature methods.
- Added supplementary material in response to reviewer 2 that illustrates the two following points
    (1) Application of QuaSoARe to flux functions that are not Lipschitz continuous,
    (2) Use of an approximate quadrature method to estimate O.

The following two sections detail the responses to comments made by the two reviewers. Note that these responses are identical to the ones made to each reviewer individually in the discussion phase of the publication process.

**Response to reviewer 1**

Comment: "This is a very useful article reminding modellers of the numerical issues that need to be considered. It should be of interest to all modellers. Many non-linear ODEs cannot be solved analytically, particularly when combined into a system of ODE equations. It is models that use such ODEs that will benefit greatly from using QuaSoARe."

Response: We thank the reviewer for their positive comments on QuaSoARe.

Comment: "The issue with hydrological models is that typically, the time step employed is dictated by the available data, not what is needed to ensure a sufficiently accurate estimation of the solution of the ODEs included in the model. The use of a coarse time step in the input data means a loss of information about what is happening at finer time scales, leading to uncertainty in the model output."

Response: In order to precise this point, we added the following comment above equation 1:

"More precisely, let us denote the reservoir volume by $S$ and assume that the reservoir is submitted to forcing variables $\tilde{V}$ that remain constant over the time step. This

assumption corresponds to most practical hydrological modelling scenarios where the time distribution of the forcings during the time step is unknown (e.g. time average of rainfall or potential evapotranspiration)."

Comment: "In Appendix A, an alternative approach would be to take the Taylor series approximation about the centre point of the interval. This would not ensure a match at the ends of the interval, leading to a likely discontinuity between intervals that would not be desirable. The approach taken ensures a continuous approximation of the function (noting that it will be discontinuous in the first derivative) and is effective providing the function is sufficiently close to a quadratic form. This will depend on the width of the interval and the variability of the function within that interval. The illustrative example used is S3, so a quadratic approximation would work well providing that the interval is not so large that the value of S ranges from near zero to a large enough value within the interval (see discussion of the illustrative example on page 8). This then defines the acceptable interval width that should be used. In general applications, this will depend on the form of the function f."

Response: We agree with the reviewer regarding the fact that QuaSoARe performance will depend on the form of the flux functions and on the quality of the interpolation. This is why we highlighted this point in Section 4 of the paper among other limitations of the method. Based on a comment from reviewer 2, we expanded this section in order to highlight the issue of flux functions that are not Lipschitz continuous.

We hope that this section will suffice to flag potential pitfalls and their remedies when using QuaSoARe.

Comment: "Minor comments"

Response: We thank the reviewer for their careful inspection of the figures in our manuscript. All minor comments identified by the reviewer were fixed in the revised manuscript.

Comment: "Typographical/grammatical errors"

Response: We thank the reviewer for their careful inspection of our manuscript. All errors identified by the reviewer were fixed in the revised manuscript.

**Response to reviewer 2**

Comment: The paper is written in a very straightforward way and is relatively easy to follow with some familiarity in analytical and numerical integration (apart from the very first part where continuity assumptions are stated, see comments below). I am overall supportive of the approach which has a great potential for speeding up current formulations of many lumped models, but I still have questions regarding the kind of solutions which should be put forward to cure the problems raised by Clark and Kavetski about the numerical soundness of many such models, in the case of multistate model structures.

Response: We thank the reviewer for their overall support and excellent comments that have helped us clarified several important aspects of the manuscript.

Comment: I think that the introduction somehow lacks a brief presentation of the context in which unbalanced, cumulative outputs from model reservoirs could happen.

Response: We agree with the reviewer and added the following two sentences in the first paragraph:

"Such reservoirs are extensively used in rainfall-runoff models such as GR4J (Perrin et al., 2003), HBV (Bergstrom and Forsman, 1973), IHACRES (Croke and Jakeman, 2004) and SAC-SMA (Burnash and Ferral, 1981) where the reservoir dynamic is described by a differential equation relating the change in storage with inputs and outputs fluxes. If the model is applied to a time step long enough for storage to change significantly, this equation must be integrated to obtain the storage level at the end of the time step and the flux totals. However, apart from a few simple cases, there is no analytical solution to this mathematical problem, and one has to revert to numerical approximations (Clark and Kavetski, 2010; Kavetski and Clark, 2010). Furthermore, flux functions in hydrological models are often highly non-linear, which magnifies numerical errors when using inappropriate numerical schemes (Kavetski and Clark, 2011). This, in turn, degrades model simulation and calibration due to the extra-parameterisation needed to compensate for these errors (Kavetski et al., 2006)."

Comment: Restricting the analysis to such autonomous ODEs (Clark and Kavetski, 2010) is very common and is a lesser loss of generality than assumption (i), but it is still worth mentioning.

Response: We agree with the reviewer and added the following clarification above Equation 1:

"More precisely, let us denote the reservoir volume by $S$ and assume that the reservoir is submitted to forcing variables $\tilde{V}$ that remain constant over the time step. This assumption corresponds to most practical hydrological modelling scenarios where the time distribution of the forcings during the time step is unknown (e.g. average rainfall or potential evapotranspiration)."

Comment: My first question would be: in validating model outputs against observations (e.g. runoff, actual evapotranspiration, recharge, etc.), why wouln't Eq. 3 be suitable for *flux* comparison, and why do we absolutely need to compute integrated quantities such as the $O_i$s (runoff *depth* over $\delta$, evapotranspiration *depth*, recharge *depth*, etc.) which have dimension L (length), rather than to their instantaneous counterparts  (runoff *rate*, evapotranspiration *rate*, etc. with dimension $L \cdot T^{-1}$) at successive times $t, t + \delta, t + 2\delta$, etc.

Response: We agree with the reviewer that it is possible to use an approximation of Eq 2 to compute the flux terms Oi. We thank the reviewer for flagging this critical point which we have now clarified by adding the following sentence above Eq 3:

"In addition, aside of applying the right ODE integrator to Eq. 1, solving the reservoir equation requires a numerical integration of Eq. 2 using a potentially different algorithm. For example, a simple quadrature method can be used to estimate the integral in the right-hand-side of Eq. 2 based on the two values S0 and S($\delta$). However, this approach can be highly inaccurate if the solution s(t) does not vary linearly with time as demonstrated in the supplementary material. A more accurate method is to expand Eq1 into a system of differential equations by adding one scalar differential equation for each flux."

In addition, we added a section in the supplementary material to show a comparison between computing the flux from a quadrature approach, as suggested by the reviewer, and by integrating the extended ODE as initially indicated in our paper. This comparison suggests that the quadrature approach is most of the time accurate but can generate large errors if the solution s(t) varies non-linearly over the time step (e.g., during high flow regime).

Comment: Clearly, solving flux equations individually is a *sufficient* condition to fulfill the conditions (B), but I rather disagree on the fact that it is a *necessary* one, as stated in lines 60–70 ("*Finally, jointly solving Eq. 1 and Eq. 2 using a numerical solver requires transforming the scalar equation Eq. 1 to a system of differential equations by adding one scalar equation for each flux*"). If we have a procedure to solve the single, state-space formulated ODE in $S$, attributing residual errors in cumulative balance to any of

the outputs would yield the same result at lower computational cost. If we chose an output $O_{i0}$ for balance error compensation, there are several possible fixes allowing to fullfill conditions (B):

Response: See previous response.

Comment: Using the language of atmospheric modeling we might say that the $O_i$'s are some kind of *diagnostic* variables that could be computed at the end of the simulated time period, $S$ being the only variable truly belonging to the category of *state variables*. Adding this distinction in the formulation of the problem could improve the paper and help modelers think about the way their models are written (either discrete time, or state-space).

Response: We agree with the reviewer and thank them for this very useful comment. We incorporated the term "diagnostic variable" in a sentence at the end of section 1.1:

"It is important to note that the computation of Oi does not affect the solution S(t). To follow the terminology of Atmospheric Modelling, Oi is a diagnostic variable (American Meteorological Society, 2024) whereas S is a prognostic variable."

Comment: This would be my main question about a possible extension of QuaSoARe to the case of multistate models: can we avoid operator splitting, and if no, shouldn't we still favor direct integration of the full, coupled system of ODEs describing the evolution of the state *vector* $\mathbf{S} \in \mathbb{R}^m$ using the semi-implicit Euler (SIE) scheme for example?

Response: The starting point of our paper, stated in the second paragraph of Section 1.1, is that direct integration of coupled reservoir systems using modern ODE integrators has not been adopted widely in hydrology. In this paragraph and in Section 1.2, we suggest that this is due to the complexity of the existing numerical methods and the additional burden they add to a model code. Motivated by this unsatisfactory situation, we developed QuaSoARe while making clear (in the abstract, at the end of the first and second paragraphs of section 1.1, at the beginning of section 4 and in the conclusion) that our method is limited to solving a scalar ODE.

We also acknowledged in the first sentence of Section 4, that it is preferrable to solve multiple stores jointly, hence following the reviewer suggestion. To make this point clear, we reworded the second sentence of Section 4 as follows:

"This is an important limitation of QuaSoARe as most hydrological models contain multiple stores that could benefit from a joint solution using a Runge–Kutta method, as presented by Clark and Kavetski (2010)."

Comment: The introduction of the paper quickly gets us into the swing of things with the mathematical assumptions stated p.2 (l. 36–41). It is a bit difficult to understand the practical implications of the Lipschitz-continuity assumption: does it mean that some functions are not admissible as flux functions?

Response: The reviewer is correct in stating that certain flux functions are not admissible as per our Lipschitz-continuity assumption. To clarify this point, we added the following paragraph in Section 4:

"Another limitation of QuaSoARe comes from the mathematical assumptions introduced in Section 1, and more specifically the need for flux functions to be Lipschitz continuous, which is equivalent to having a bounded derivative if the function is absolutely continuous. This assumption is required to ensure the unicity of the solutions of Eq. 1, and hence its monotonous nature as demonstrated at the beginning of Section 2.2 but eliminates many common flux functions encountered in hydrological systems (for example power functions of S with an exponent lower than 1). A first solution to this problem is to alter the flux functions to obtain smoother functions with bounded derivatives following Kavetski et al. (2006, see Section 5 of this paper) and, hence, revert to the domain of applicability of QuaSoARe. If this is not an option, QuaSoARe may still generate reasonable simulations if the solution of Eq. 1 does not come close to any discontinuity in the flux functions derivatives. This is, of course, case specific and subtle because it goes beyond the theoretical validity of QuaSoARe. An example of such a case is presented in supplementary material."

Comment: If we consider the classical equation of a draining tank, (...) This function is not Lipschitz-continuous on the possible range of storage values (...). Does it mean that we have to check each flux function?

Response: The reviewer is correct is saying that all functions need to be checked for Lipschitz-continuity. However, as said in the response above, we believe that QuaSoARe could still generate reasonable predictions if the solution s(t) is not approaching the points where the derivative of the flux functions becomes unbounded. To illustrate this point we added an example in the supplementary material.